# CACE-NET: CASCADE COUPLING EFFECT FOR LINK PREDICTION IN MULTI-LAYER NETWORKS

## ABSTRACT

As social infrastructure networks become increasingly complex, the interdependence between different network layers has garnered significant attention. In real-world multi-layer networks, structural changes in one layer often trigger cascade coupling effects, where these changes propagate across layers and influence link formation in a chain-like manner. However, traditional link prediction methods typically treat each layer independently, overlooking these cross-layer dependencies. To address this, we propose CACE-NET: **CA**scade **C**oupling **E**ffect for link prediction in multi-layer **NET**works. CACE-NET encomasses three key components: (i) Layer-wise representation extractor; (ii) Adversarial coupling representation encoder; and (iii) Adaptive fusion link predictor. Firstly, layer-wise representation extractor applies independent graph convolutions to model intra-layer structures. Next, the adversarial coupling representation encoder leverages adversarial training to learn latent cascading dependencies between replica nodes across layers. Finally, adaptive fusion predictor integrates intra-layer and cross-layer embeddings via an attention mechanism, effectively combining local and global information to enhance link prediction in the target layer. Experimental results on multiple real-world datasets show that CACE-NET outperforms state-of-the-art methods, achieving AUC improvements of up to 13.29%.

## 1 INTRODUCTION

As real-world systems become increasingly interconnected, many complex relationships can be effectively modeled as multi-layer networks, including power grids, transportation systems, and social infrastructure networks (Hammoud & Kramer, 2020). A multi-layer network models complex systems by organizing nodes and edges into multiple layers, each representing a different type of relationship among the same entities (Boccaletti et al., 2014; Kivelä et al., 2014). Each entity *"physical node"* can appear as a *"replica"* in different layers, participating in layer-specific interactions. Inter-layer edges capture dependencies between replicas across layers, allowing multi-layer networks to represent the diverse and interconnected relationships (Jiang et al., 2020; De Domenico, 2023). These cross-layer dependencies can manifest as cascade coupling effects, where variations in one layer trigger a chain reaction across others (Danziger & Barabási, 2022).

Such cascade dynamics make link prediction particularly challenging, as changes in one layer can propagate through replica nodes and reshape connections in others. As shown in Fig. 1, a new connection in the *"Workplace"* layer (where the *"Engineer"* links the *"Manager"* and *"Researcher"*) can lead to new friendships, which in turn facilitate collaborations in the *"Project"* layer. By capturing such cross-layer cascade signals, the models may leverage historical interactions from multiple layers to improve link prediction in the target layer. Thus, explicitly modeling these cross-layer coupling effects is essential for accurate and reliable link prediction in multi-layer networks.

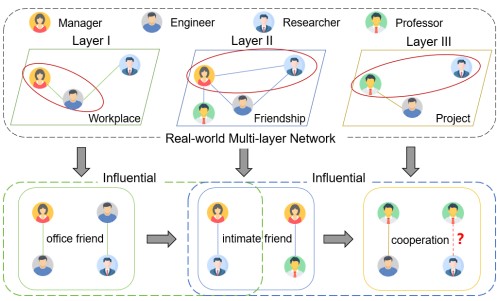

Figure 1: Illustration of inter-layer coupling effects in multi-layer networks for link prediction.

Most existing approaches lack explicit mechanisms to capture cascading coupling effects, limiting their effectiveness in complex multi-layer networks. We group current link prediction methods into three main categories: **(a) Network simplification methods**, which aggregate all layers into a single weighted network (Mishra et al., 2023). While this simplifies analysis, it discards nuanced inter-layer interactions and overlooks cascading effects from structural changes in one layer to others. **(b) Inter-layer similarity-based methods**, which use predefined similarity metrics to compare node or edge attributes across layers (Yao et al., 2017; Najari et al., 2019). These methods offer interpretability and highlight some cross-layer patterns, but rely on static calculations and cannot model the dynamic, sequential propagation of changes across layers. **(c) Extended cross-layer embedding methods**, which align entities via replica nodes and extend GCN operations to aggregate information from both intra-layer neighbors and cross-layer counterparts (Ren et al., 2024). However, these approaches depend on static coupling assumptions and direct replica connections, limiting their ability to capture latent, long-range, and sequential dependencies underlying cascading coupling effects in real-world networks.

While recent progress has been made in link prediction for multi-layer networks, most existing methods still fall short of explicitly modeling the cascading coupling effects that are fundamental to real-world systems. This oversight leaves a significant gap: the dynamic, sequential propagation of structural changes across layers—and its impact on link formation—remains insufficiently addressed. In this work, we propose a novel framework that explicitly models cascading inter-layer dependencies for improved link prediction. Unlike previous methods that rely on static feature aggregation (Wang & Sun, 2021), our approach introduces a dynamic and adaptive mechanism to capture inter-layer coupling, enabling the model to learn robust and transferable representations that reflect the evolving relationships in multi-layer networks. Our work is motivated by two central challenges: _First_, how to effectively extract meaningful coupling information from node replicas distributed across different layers, while mitigating the risk of information redundancy that can arise from their presence. _Second_, how to dynamically balance intra-layer structural information with cross-layer coupling signals during prediction.

To tackle these challenges, we introduce CACE-NET: **CA**scade **C**oupling **E**ffect for link prediction in multi-layer **NET**works, a framework that explicitly models dynamic inter-layer coupling effects for link prediction in multi-layer networks. Rather than treating each layer in isolation or relying on static aggregation, CACE-NET leverages adversarial training to capture latent dependencies among replica nodes across layers. The framework consists of three core components: (i) Layer-wise representation extractor that applies independent graph convolutions to encode intra-layer structures; (ii) Adversarial coupling representation encoder that employs adversarial learning to uncover and distill cross-layer coupling information among node replicas; and (iii) Adaptive fusion link predictor that adaptively integrates intra-layer and cross-layer representations via an attention mechanism, enabling accurate link prediction in the target layer by balancing local and global information.

We summarize the key contributions of this work as follows:

- We introduce CACE-NET, a novel framework for link prediction in multi-layer networks that explicitly models both intra-layer structure and dynamic inter-layer coupling effects. By leveraging adversarial training, our coupled information encoder learns robust, shared cross-layer representations, while an adaptive fusion mechanism integrates these with intra-layer features to produce informative, stable embeddings for prediction.
- We provide formal analysis establishing that CACE-NET: (i) learns stable and robust cross-layer embeddings under bounded adversarial perturbations (Proposition 1); (ii) fuses intra- and cross-layer information via a Lipschitz attention module controlling perturbation propagation (Lemma 1); and (iii) achieves end-to-end prediction stability, with sensitivity bounded by a constant and controlled by learning rate and adversarial step size (Lemma 2).
- Extensive experiments on real-world datasets demonstrate that CACE-NET consistently outperforms state-of-the-art baselines, achieving up to 13.29% higher AUC and validating the benefits of explicitly modeling cascading coupling effects.

## 2 RELATED WORK

With the deepening of research on social networks, an increasing number of scholars are focusing on multi-layer social networks. Recent studies demonstrate a certain degree of correlation between the topological features of different layers in multi-layer networks (Szell et al., 2010; Lee et al., 2015).

Topology-based methods primarily rely on the structural properties of the network to perform link prediction. These methods compute the similarity between two nodes using topological features such as common neighbors, path-based information, and others. For instance, Yao et al. (2017) propose a method that calculates node similarity by combining both intra-layer and inter-layer topological information. Similarly, Najari et al. (2019) incorporate inter-layer similarity along with target-layer features to enhance prediction accuracy. These methods argue that if two layers are similar, then links in one layer are more likely to appear in the other layer.

Some studies simplify the problem by aggregating multi-layer networks into a single-layer representation. For example, Mishra et al. (2022) integrate all layers into a weighted static network, employing node and edge relevance for link prediction. Likewise, Mishra et al. (2023) introduce long-range path information and iteratively compute link likelihoods.

Feature learning-based methods use machine learning or deep learning to learn representations from the network structure for link prediction. Some approaches adopt classical machine learning algorithms (Jalili et al., 2017), while others improve deep models through careful feature selection (Mandal et al., 2018). More recent works focus on extracting elaborate structural representations from all layers (Shan et al., 2020), reconstructing the target layer by leveraging cross-layer representations (Abdolhosseini-Qomi et al., 2020), or treating the fusion of multi-layer network information as a multi-attribute decision-making process to enhance predictive accuracy (Luo et al., 2021).

Although prior work advanced multi-layer link prediction via topology, feature learning, and aggregation, these methods often overlook the intricate and dynamic inter-layer dependencies that drive link formation. In contrast, our method directly addresses this limitation by explicitly modeling cascade coupling effects, leveraging adversarial training and adaptive attention mechanisms. By capturing how structural changes in one layer can propagate and influence link formation in others, our approach enables a deeper understanding of cross-layer interactions and achieves superior prediction performance in complex multi-layer network settings.

## 3 PRELIMINARIES

In this section, we formally define the problem of link prediction in multi-layer networks. For a detailed overview of the foundational concepts and methods employed in this paper—including Graph Convolutional Networks (GCNs), FreeLB adversarial training, and the formal definition of coupling in multi-layer networks—please refer to Appendix A.

**The Problem.** Let $G = \{G_1, G_2, \ldots, G_K\}$ denote a multi-layer network with $K$ layers, where each layer $G_i = (V_i, E_i)$ consists of a node set $V_i$ and an edge set $E_i$. All layers share a common set of underlying entities $V$. However, not every entity necessarily appears in all layers. For layer $G_i$, its node set can be formalized as $V_i = \{\, u_i \mid u \in V_i \subseteq V \,\}$, where $u_i$ denotes the replica of entity $u$ in the $i$-th layer. To capture the cross-layer consistency, we define a mapping $\pi(u_i) = u$, indicating that all replica nodes correspond to the same physical entity. In this way, the coupling among layers reflects a *partial alignment*: some entities have replicas in all layers, while others only appear in a subset of layers. The cascade coupling effect refers to the phenomenon that information can propagate across layers through replica nodes, even when some entities are missing in certain layers. Specifically, if an entity $u$ is absent in layer $G_1$ but present in another layer $G_2$, its replica $u_2$ can still serve as an intermediary to transmit information back to $G_1$ through cross-layer coupling. In this way, missing structural or relational information in one layer can be compensated by the presence of the same entity in other layers, forming a cascade of cross-layer influences that enhances robustness and expressiveness in multi-layer networks.

Then let $X \in \mathbb{R}^{|V| \times d}$ denote the feature matrix of all node entities, where $d$ is the feature dimension and $X_u$ is the feature vector of node $u \in V$. The intra-layer node embeddings for all layers are denoted by $H = \{H_1, H_2, \ldots, H_K\}$, where $H_i \in \mathbb{R}^{|V| \times h}$ is the embedding matrix for layer $i$ and $h$ is the embedding dimension. We designate one layer $G_t$ as the *target layer*, and define the set of *auxiliary layers* as $G_A = \{G_a \mid a \in \{1, 2, \ldots, K\}, a \neq t\}$. By construction, $G_t \cap G_A = \emptyset$ and $G_t \cup G_A = G$. The link prediction function is denoted as $\hat{S}(u, v)$, which estimates the probability that a link exists between nodes $u$ and $v$ in the target layer $G_t$. The edge set of the target layer $E_t$ is partitioned into disjoint subsets: training edges $E_{\text{train}}$, validation edges $E_{\text{val}}$, and test edges $E_{\text{test}}$, *i.e.*, $E_t = E_{\text{train}} \cup E_{\text{val}} \cup E_{\text{test}}$ and $E_{\text{train}} \cap E_{\text{val}} \cap E_{\text{test}} = \emptyset$. In contrast, the edge sets of auxiliary layers are entirely used for training as supplementary information.

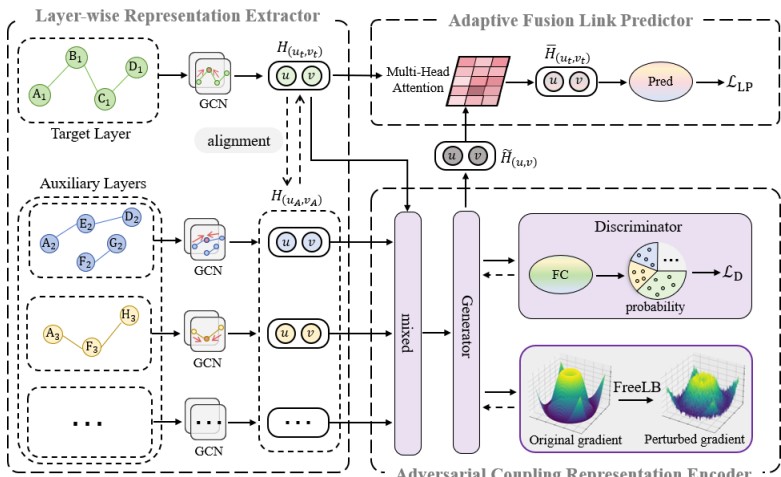

Figure 2: Architecture of CACE-NET.

# 4 CACE-NET

**Overview.** Fig. 2 illustrates CACE-NET, comprising three modules: (i) **Layer-wise Representation Extractor** — independent GCNs per layer to encode intra-layer structure; (ii) **Adversarial Coupling Representation Encoder** — a generator–discriminator that learns layer-invariant cross-layer embeddings using FreeLB perturbations; and (iii) **Adaptive Fusion Link Predictor** — a multi-head attention module that fuses intra-layer and cross-layer embeddings for link prediction. Together, these modules integrate structural and coupling signals for accurate link prediction in multi-layer networks. Further details about each section are as follows.

## 4.1 LAYER-WISE REPRESENTATION EXTRACTOR

The layer-wise representation extractor is designed to independently learn the structural features of each layer in a multi-layer network, focusing exclusively on intra-layer topology. Formally, given a multi-layer network where each layer $G_i = (V_i, E_i)$ comprises a node set $V_i$ and an edge set $E_i$, we assume $V_i \subseteq V$, where $V$ denotes the global set of physical nodes shared across all layers. To better account for cross-layer coupling, we augment each layer's node set to the full $V$ by adding isolated nodes for entities not originally present in that layer, ensuring that every physical node $u \in V$ has a corresponding replica $u_i$ in each layer $G_i$. We then assign a dedicated GCN for each layer, ensuring that the unique local patterns of each layer are captured without interference from other layers.

Specifically, for each layer $G_i$, we compute node embeddings using a 2-layered GCN, following standard practice (Kipf & Welling, 2016). The GCN iteratively aggregates information from each node's local neighborhood, enabling the model to encode the structural context of each node within its respective layer. While all layers utilize the same GCN architecture, each layer maintains its own set of learnable parameters (i.e., weight matrices and biases). This design allows the extractor to model layer-specific structural characteristics. We define the layer-wise representation as follows:

$$H = \{H_i\}_{i=1}^K = \{\text{GCN}_i(G_i, A_i)\}_{i=1}^K, \tag{1}$$

where $A_i$ is the adjacency matrix of layer $G_i$, and each $\text{GCN}_i$ is an independent encoder dedicated to extracting node representations for its respective layer. The embedding set $\{H_i\}_{i=1}^K$ covers all $K$ layers. For a physical node $u$ shared across layers, we denote its replicas as $u_1, \ldots, u_K$, with $u_i$ representing $u$ in layer $G_i$. Their intra-layer embeddings $H_{u_1}, \ldots, H_{u_K}$ capture the layer-specific structural features of $u$. For each link $(u_i, v_i)$ in layer $G_i$, we construct its intra-layer embedding by vector concatenation $\|$ of the embeddings of the endpoints:

$$H_{(u_i, v_i)} = H_{u_i} \| H_{v_i}. \tag{2}$$

For link prediction in the target layer $G_t$, we utilize both its intra-layer information and that from auxiliary layers $G_A = \{G_a \mid a \neq t\}$. For a node pair $(u_t, v_t)$ in $G_t$, we collect its replicas $(u_a, v_a)$ in each auxiliary layer $G_a$. The intra-layer embeddings $H_{(u_t, v_t)}$ (target) and $H_{(u_a, v_a)}$ (auxiliary) together form the set $\{H_{(u_i, v_i)}\}_{i=1}^K$, capturing structural features across all layers. These embeddings serve as the basis for subsequent cross-layer coupling and fusion, enabling the model to integrate both local and inter-layer information for improved link prediction.

## 4.2 ADVERSARIAL COUPLING REPRESENTATION ENCODER

To model cross-layer dependencies, we propose an adversarial coupling encoder using a generator-discriminator framework. This module contains a generator producing cross-layer embeddings for node pairs, and a discriminator identifying each embedding's layer origin. The generator is trained to learn latent coupling relationships between corresponding nodes across layers, aiming to produce embeddings that are indistinguishable to the discriminator. In contrast, the discriminator is optimized to correctly classify the layer of origin for each embedding. This adversarial interplay encourages the generator to distill shared, layer-invariant features while suppressing irrelevant, layer-specific noise. The resulting robust embeddings capture deep structural patterns and latent couplings, enhancing cross-layer generalization in multi-layer networks.Further details are as follows.

**Generator.** The generator is designed to produce cross-layer coupling embeddings for node pairs by refining their representations from each layer. For a given node pair $(u, v)$, let $H_{(u_t, v_t)}$ denote its embedding in the target layer and $H_{(u_a, v_a)}$ its embedding in each auxiliary layer $G_a$. The generator takes these embeddings and aims to extract a shared, layer-invariant representation. To achieve this, we apply adversarial optimization by introducing a bounded perturbation to the input embeddings before passing them through the transformation function $f_\theta$.

Specifically, we adopt the FreeLB adversarial training approach Zhu et al. (2020), which iteratively applies gradient-based perturbations to the embeddings over multiple steps. This process generates adversarial examples that are smoother and more robust, helping the generator learn representations resilient to structural variations across layers. The update rule for the perturbation is given by:

$$\delta^{(t+1)} = \Pi_{\|\delta\| \leq \epsilon} \left[ \delta^{(t)} + \alpha \cdot \text{sign} \left( \nabla_{\delta^{(t)}} \mathcal{L}_D(f_\theta(H + \delta^{(t)})) \right) \right], \tag{3}$$

where $\mathcal{L}_D$ is the adversarial loss (cross-entropy, see Equation 7), $\epsilon$ is the maximum allowed perturbation norm, $\alpha$ is the step size, $\delta$ is the adversarial perturbation, and $\Pi$ projects $\delta$ onto the $\ell_\infty$-ball of radius $\epsilon$. Repeating the process for $T$ steps produces the final perturbed embedding:

$$\widetilde{H}_{(u,v)} = f_\theta(H_{(u_i, v_i)} + \delta^{(T)}), \tag{4}$$

where $f_\theta$ is implemented as a multi-layer perceptron (MLP) that maps the perturbed node-pair embeddings into a shared latent space. This shared space is designed to capture cascade coupling messages $\widetilde{H}_{(u,v)}$, *i.e.*, the cross-layer information that propagates among replica nodes. In this way, missing or sparse structural patterns in the target layer can be compensated by auxiliary layers through cross-layer coupling. The enriched representations are then leveraged for more accurate link prediction in the target layer, improving overall performance in multi-layer networks.

Below, we formalize a proposition that emphasizes the stability and robustness of our adversarial coupling encoder. Specifically, when the generator and discriminator are Lipschitz and adversarial perturbations are bounded (as in FreeLB), the min–max training dynamics ensure that both the parameters and the resulting cross-layer embeddings remain within a bounded, stable region.

**Proposition 1 (Robust coupling under bounded perturbations).** Assume the generator $f_\theta$ and discriminator $D_\phi$ are Lipschitz and FreeLB uses $T$ steps with step size $\alpha$ and budget $\epsilon$. Then generator outputs remain in a bounded set and the sequence of embeddings $\{\widetilde{H}_{(u,v)}\}$ is stable under the min–max game in Equation 8.

A comprehensive theoretical justification for Proposition 1 is presented in Appendix C.1.

**Discriminator.** The discriminator is a fully connected neural network that identifies the originating layer of each embedding from the generator. By classifying the layer index of a given embedding $\widetilde{H}$, the discriminator pushes the generator to produce representations that are indistinguishable across layers, promoting the learning of layer-invariant, transferable features.

Specifically, given a node pair $(u, v)$, we obtain $K$ embeddings—one from each layer. The discriminator, $D_\phi$, is a multi-class classifier parameterized by $\phi$ and trained to predict the correct layer label for each embedding. Formally, $D_\phi$ is defined as:

$$D_\phi(\widetilde{H}) = \text{softmax}(W_\phi \widetilde{H} + b_\phi), \tag{5}$$

where $W_\phi \in \mathbb{R}^{K \times d}$ is a learnable weight matrix, $b_\phi \in \mathbb{R}^K$ is a bias vector, and $d$ is the dimension of the input embedding. The $i$-th entry of the output, $D_\phi(\widetilde{H})_i$, represents the predicted probability

that $\widetilde{H}$ originates from the $i$-th layer:

$$D_\phi(\widetilde{H})_i = \frac{\exp\left((W_\phi\widetilde{H} + b_\phi)_i\right)}{\sum_{j=1}^{K}\exp\left((W_\phi\widetilde{H} + b_\phi)_j\right)}. \tag{6}$$

To train the discriminator to accurately identify the originating layer of each embedding, we employ the standard cross-entropy loss:

$$\mathcal{L}_D = -\mathbb{E}_{\widetilde{H}_{(u,v)}\sim E_{\text{train}}}\log D_\phi(\widetilde{H}_{(u,v)})_{y_{(u,v)}}, \tag{7}$$

where $y_{(u,v)}$ denotes the true layer label for the embedding $\widetilde{H}_{(u,v)}$, and $D_\phi(\widetilde{H}_{(u,v)})_{y_{(u,v)}}$ is the predicted probability assigned to the correct layer. This objective encourages the discriminator to maximize the likelihood of correctly classifying the layer of origin for each embedding, thereby sharpening its ability to distinguish between representations from different layers.

**Adversarial Training Process.** For adversarial training, we employ a min-max optimization: the generator seeks to produce cross-layer embeddings that fool the discriminator, while the discriminator tries to correctly identify each embedding's layer of origin. The adversarial objective is:

$$\min_\theta \max_\phi \mathbb{E}_{\widetilde{H}_{(u,v)}\sim E_{train}} \sum_{i=1}^{K} \mathbb{1}[y_{(u,v)} = i]\log D_\phi(\widetilde{H}_{(u,v)})_i. \tag{8}$$

The training alternates between updating $D_\phi$ to improve layer classification and updating $f_\theta$ to make embeddings more layer-invariant. This process continues until the discriminator can no longer distinguish layers, indicating the generator has learned robust, shared cross-layer representations that enhance link prediction.

### 4.3 ADAPTIVE FUSION LINK PREDICTOR

After obtaining both the intra-layer embeddings and the cross-layer coupling embeddings, the Adaptive Fusion Link Predictor integrates these representations to perform link prediction in the target layer $G_t$. For each node pair $(u, v)$, we combine its intra-layer embedding $H_{(u_t,v_t)}$ and cross-layer embedding $\widetilde{H}_{(u,v)}$ using a multi-head attention mechanism. This mechanism enables the model to adaptively weigh and fuse information from both sources. The attention head is defined as follows:

$$\text{Head}_i = \texttt{Attention}(Q_i, K_i, V_i) = \texttt{softmax}\left(\frac{Q_i K_i^\top}{\sqrt{d_k}}\right)V_i, \tag{9}$$

where $\text{Head}_i$ represents the output of the $i$-th attention head, $d_k$ is the dimension of the key vectors, $W_i^Q, W_i^K, W_i^V$ are trainable weight matrices:

$$Q_i = W_i^Q[H_{(u_t,v_t)} \parallel \widetilde{H}_{(u,v)}], K_i = W_i^K[H_{(u_t,v_t)} \parallel \widetilde{H}_{(u,v)}], V_i = W_i^V[H_{(u_t,v_t)} \parallel \widetilde{H}_{(u,v)}], \tag{10}$$

where $\parallel$ denotes the concatenation of $H_{(u_t,v_t)}$ and $\widetilde{H}_{(u,v)}$. The final fused embedding is:

$$\overline{H}_{(u_t,v_t)} = \texttt{concat}(\text{Head}_1, \ldots, \text{Head}_h). \tag{11}$$

Building on Proposition 1, Lemma 1 establishes that the downstream fusion module is Lipschitz continuous, ensuring that small perturbations in $[H_{(u_t,v_t)} \parallel \widetilde{H}_{(u,v)}]$ are not amplified. As a result, the learned coupling embeddings $\widetilde{H}_{(u,v)}$ exhibit robustness to adversarial noise and stochastic fluctuations, and their influence on the final prediction remains well-controlled.

**Lemma 1 (Lipschitz fusion).** Let $\|W_i^Q\|, \|W_i^K\|, \|W_i^V\| \leq c$ for all heads and use scaled dot-product attention. Then there exists $C = C(c, h)$ such that, for $x = [H_{(u_t,v_t)} \parallel \widetilde{H}_{(u,v)}]$,

$$\|\overline{H}_{(u_t,v_t)}(x) - \overline{H}_{(u_t,v_t)}(x')\| \leq C\|x - x'\|.$$

This implies the fusion does not amplify small embedding perturbations.
A comprehensive theoretical justification for Lemma 1 is presented in Appendix C.2.

The final fused embedding $\overline{H}_{(u_t,v_t)}$ serves as the input for predicting the likelihood that a link exists between nodes $u$ and $v$ in the target layer $G_t$:

$$P_{(u_t,v_t)} = \sigma(w^T f(\overline{H}_{(u_t,v_t)})), \tag{12}$$

where $\sigma(\cdot)$ denotes the sigmoid activation function, $w$ is a trainable weight vector, and $f(\cdot)$ represents a multi-layer perceptron (MLP). The model is trained by minimizing the binary cross-entropy loss $\mathcal{L}_{LP}$, as shown below:

$$\mathcal{L}_{LP} = -\mathbb{E}_{(u_t,v_t) \sim E_{train}}[y_{(u_t,v_t)} \log P_{(u_t,v_t)} + (1 - y_{(u_t,v_t)}) \log(1 - P_{(u_t,v_t)})], \tag{13}$$

where $y_{(u_t,v_t)}$ indicates whether a link exists between $u$ and $v$ in $G_t$ (1 if present, 0 otherwise). The adaptive fusion mechanism integrates intra-layer and cross-layer coupling embeddings via attention, allowing the model to dynamically weigh their contributions and optimally leverage cross-layer coupling information for improved link prediction.

### 4.4 Model Integration

CACE-Net is trained end-to-end, jointly optimizing all three modules via backpropagation. The layer-wise representation extractor generates intra-layer embeddings, the adversarial coupling encoder learns cross-layer embeddings using a generator-discriminator framework, and the adaptive fusion link predictor combines these via attention to yield the final prediction. The overall loss is:

$$\mathcal{L} = \lambda \mathcal{L}_D + (1 - \lambda)\mathcal{L}_{LP}, \tag{14}$$

where $\mathcal{L}_{LP}$ is the link prediction loss, $\mathcal{L}_D$ is the adversarial training loss and $\lambda \in [0, 1]$ weights the relative importance of adversarial training versus link prediction.

The prediction pipeline is robust: Proposition 1, Lemma 1, and the classifier's Lipschitz property ensure that small input perturbations cause only minor, controlled changes in predicted probabilities. Below, we show that stability holds with suitable learning rates and adversarial step sizes.

**Lemma 2 (Prediction stability).** Let $\sigma$ and the MLP $f$ in Equation 12 be $L_\sigma$- and $L_f$-Lipschitz, respectively. Under Proposition 1 and Lemma 1, there exists $K > 0$ such that for any two inputs $x = [H_{(u_t,v_t)} \| \widetilde{H}_{(u,v)}]$ and $x'$,

$$\left| P_{(u_t,v_t)}(x) - P_{(u_t,v_t)}(x') \right| \le K \|x - x'\|. \tag{15}$$

In particular, across successive epochs,

$$\left| P^{t+1}_{(u_t,v_t)} - P^t_{(u_t,v_t)} \right| \le K \left( c_\theta \, \eta + c_\delta \, \alpha \right), \tag{16}$$

so predictions are stable for sufficiently small learning rates $\eta$ and adversarial step sizes $\alpha$.
A comprehensive theoretical justification for Lemma 2 is presented in Appendix C.3.

**Training Workflow.** The complete training workflow for CACE-Net is summarized in Appendix Algorithm 1 in Appendix B. Given a multi-layer network $G$, the model sequentially: (1) extracts intra-layer embeddings $H$; (2) generates cross-layer embeddings $\widetilde{H}$ via the adversarial coupling representation encoder; (3) fuses them into $\overline{H}$ using the adaptive fusion link predictor; (4) computes losses $\mathcal{L}_D$, $\mathcal{L}_{LP}$, and total loss $\mathcal{L}$; (5) applies adversarial perturbations for robustness; (6) updates all parameters via backpropagation; and (7) predicts the probability of unobserved links in the target layer. For details, the complexity analysis of CACE-Net is reported separately in Appendix C.4.

## 5 Experimentation

In this section, we present the experimental results of CACE-Net on several real-world multi-layer networks. We first describe the experimental settings and then present the main results and ablation study. Owing to space constraints, additional experimental results, including Generalization Analysis, Parameter Analysis and Case Study are reported in Appendix E.

### 5.1 Experimental Settings

**Datasets.** To evaluate the performance of CACE-Net fairly, we use several real-world multi-layer networks. These include: (i) Aarhus Magnani et al. (2013); (ii) Enron Tang et al. (2012); (iii) Kapferer De Domenico et al. (2014); (iv) London De Domenico et al. (2014); and (v) Reddit Kumar et al. (2018). Further details and statistics of datasets are summarized in Appendix D.1 and Table 3.
**Baselines.** To assess the performance of our proposed CACE-Net, we compare it with several state-of-the-art methods, encompassing both traditional single-layer network approaches applied directly to the target layer, as well as specialized baseline methods designed for multi-layer networks. These include: (i) CN Kossinets (2006); (ii) Jaccard Jaccard (1901); (iii) NSILR Yao et al. (2017); (iv) SEAL Zhang & Chen (2018); (v) MultiSup Shan et al. (2020); (vi) MADM Luo et al. (2021);

Table 1: Comparison of AUC and Accuracy(Acc) across different models on real-world datasets.

| Dataset | Metric | Layer | CN | Jaccard | NSILR | SEAL | MultiSup | MADM | MNERLP | HOPLP | LUSTER | CACE-Net | Improve(%) |
|---|---|---|---|---|---|---|---|---|---|---|---|---|---|
| Aarhus | AUC | 1 | 0.8165 | 0.8278 | 0.9086 | 0.4996 | 0.8754 | 0.8714 | 0.9121 | 0.9105 | 0.9316 | **0.9627** | 3.34% |
| | | 2 | 0.6872 | 0.7448 | 0.9018 | 0.5616 | 0.8166 | 0.8636 | **0.9064** | 0.9003 | 0.8459 | 0.8114 | – |
| | | 3 | 0.6250 | 0.6250 | 0.8939 | 0.6250 | 0.7500 | 0.9113 | 0.6694 | 0.6029 | 0.9289 | **0.9866** | 6.21% |
| | | 4 | 0.6955 | 0.7405 | 0.8652 | 0.5397 | 0.7750 | 0.8749 | 0.7869 | 0.7583 | 0.8942 | **0.9328** | 4.32% |
| | | 5 | 0.7082 | 0.7179 | 0.8965 | 0.5115 | 0.8245 | 0.8303 | 0.8409 | 0.8334 | 0.9211 | **0.9537** | 3.54% |
| | Acc | 1 | 0.7051 | 0.7308 | 0.8651 | 0.4871 | 0.8750 | 0.8847 | 0.8746 | 0.8724 | 0.8587 | **0.9143** | 3.35% |
| | | 2 | 0.5800 | 0.6600 | 0.8488 | 0.5600 | 0.8235 | 0.7624 | **0.8608** | 0.8180 | 0.7533 | 0.7133 | – |
| | | 3 | 0.6250 | 0.6250 | 0.7875 | 0.6250 | 0.6333 | 0.8780 | 0.6690 | 0.6024 | 0.8750 | **0.9732** | 10.84% |
| | | 4 | 0.5294 | 0.5882 | 0.7965 | 0.5294 | 0.6750 | 0.7929 | 0.7722 | 0.7520 | 0.8015 | **0.8844** | 10.34% |
| | | 5 | 0.6282 | 0.5513 | 0.8251 | 0.4743 | 0.8214 | 0.7285 | 0.7734 | 0.7660 | 0.8524 | **0.8857** | 3.91% |
| Enron | AUC | 1 | 0.4815 | 0.4630 | 0.5096 | 0.5322 | 0.5753 | 0.4993 | 0.4861 | 0.4925 | 0.9630 | **0.9884** | 2.64% |
| | | 2 | 0.4800 | 0.4800 | 0.5835 | 0.4936 | 0.6216 | 0.5084 | 0.4894 | 0.4932 | 0.9416 | **0.9423** | 0.07% |
| | Acc | 1 | 0.4815 | 0.4815 | 0.5074 | 0.5555 | 0.4421 | 0.4967 | 0.5861 | 0.5635 | 0.9076 | **0.9661** | 6.45% |
| | | 2 | 0.4800 | 0.4800 | 0.5346 | 0.5400 | 0.6222 | 0.5129 | 0.5525 | 0.5192 | **0.8715** | 0.7975 | - |
| Kapferer | AUC | 1 | 0.6794 | 0.5952 | 0.7720 | 0.5775 | 0.8513 | 0.8101 | 0.7502 | 0.7719 | 0.8787 | **0.9296** | 5.79% |
| | | 2 | 0.5756 | 0.6711 | 0.6886 | 0.4187 | 0.7554 | 0.6705 | 0.7627 | 0.7566 | 0.7902 | **0.8549** | 8.19% |
| | | 3 | 0.6800 | 0.6133 | 0.7638 | 0.5911 | 0.7950 | 0.7780 | 0.7032 | 0.6678 | 0.8824 | **0.9197** | 4.23% |
| | | 4 | 0.6302 | 0.7285 | 0.6646 | 0.4930 | 0.7795 | 0.6856 | 0.7216 | 0.7322 | 0.8686 | **0.9010** | 3.73% |
| | Acc | 1 | 0.5484 | 0.5484 | 0.7217 | 0.5645 | 0.7578 | 0.7193 | 0.7139 | 0.7348 | 0.8077 | **0.8504** | 5.29% |
| | | 2 | 0.6556 | 0.6000 | 0.6131 | 0.4555 | 0.6869 | 0.6344 | 0.6824 | 0.6744 | 0.6698 | **0.7594** | 10.55% |
| | | 3 | 0.7333 | 0.6667 | 0.6960 | 0.6000 | 0.7457 | 0.6871 | 0.6828 | 0.7158 | 0.7933 | **0.8558** | 7.88% |
| | | 4 | 0.5526 | 0.6053 | 0.6069 | 0.5263 | 0.6650 | 0.6610 | 0.6532 | 0.6518 | 0.7783 | **0.8261** | 6.14% |
| London | AUC | 1 | 0.5079 | 0.5556 | 0.5275 | 0.6505 | 0.6431 | 0.5534 | 0.5333 | 0.5284 | 0.8282 | **0.9383** | 13.29% |
| | | 2 | 0.4118 | 0.4706 | 0.5006 | 0.6435 | 0.7937 | 0.5689 | 0.4992 | 0.4998 | 0.8506 | **0.9237** | 8.59% |
| | | 3 | 0.5556 | 0.6111 | 0.5236 | 0.5864 | 0.7575 | 0.4987 | 0.5250 | 0.5175 | **0.9000** | 0.9368 | 4.09% |
| | Acc | 1 | 0.5079 | 0.5476 | 0.5275 | 0.6111 | 0.6326 | 0.5535 | 0.5333 | 0.5285 | 0.7627 | **0.8378** | 9.85% |
| | | 2 | 0.4118 | 0.4706 | 0.5007 | 0.5882 | 0.7250 | 0.5694 | 0.4992 | 0.4998 | 0.7997 | **0.8697** | 8.75% |
| | | 3 | 0.5556 | 0.6111 | 0.5237 | 0.5444 | 0.7166 | 0.4987 | 0.5250 | 0.5175 | 0.8360 | **0.8624** | 3.16% |
| Reddit | AUC | 1 | 0.8298 | 0.8207 | 0.8633 | 0.5238 | 0.7779 | 0.8300 | 0.8689 | 0.8593 | 0.8930 | **0.9165** | 2.63% |
| | | 2 | 0.8182 | 0.8200 | 0.8613 | 0.4724 | 0.7967 | 0.8524 | 0.8413 | 0.8295 | 0.8830 | **0.9025** | 2.21% |
| | Acc | 1 | 0.5001 | 0.5002 | 0.7310 | 0.5077 | 0.7057 | 0.5012 | 0.7139 | 0.7148 | 0.8123 | **0.8403** | 3.45% |
| | | 2 | 0.5005 | 0.5035 | 0.7369 | 0.4984 | 0.7303 | 0.5003 | 0.7255 | 0.7291 | 0.8056 | **0.8278** | 2.76% |

(vii) MNERLP Mishra et al. (2022); (viii) HOPLP Mishra et al. (2023); and (ix) LUSTER Yang et al. (2025). Further details about the baseline methods are provided in Appendix D.2.

**Evaluation Metrics.** For performance evaluation, we utilize two primary metrics, namely: Area Under the ROC Curve (AUC) and Accuracy (Acc). Further details and mathematical formulation of these metrics are provided in Appendix D.3.

**Experimental Setup.** Detailed experimental parameters, programming framework and hardware configuration are presented in Appendix D.4.

## 5.2 MAIN RESULTS

Table 1 presents a comprehensive comparison of CACE-NET with various baseline link prediction methods across multiple datasets. Overall, CACE-NET consistently outperforms the baselines, highlighting the advantage of explicitly modeling cross-layer coupling information. In contrast, most baseline methods rely solely on intra-layer structural features, which limits their ability to capture the complex dependencies present in multi-layer networks.

A closer examination reveals that the performance gains of CACE-NET are particularly pronounced on the London dataset. Specifically, for layer 1 of London, it achieves improvements of 13.29% in AUC and 9.85% in Accuracy. As shown by the structural statistics in Table 3, this layer is characterized by near-zero values of $\rho_i$ and $c_i$, reflecting extremely sparse connectivity and low clustering. In such cases, the scarcity of intra-layer information makes the integration of auxiliary cross-layer signals critical, which accounts for the substantial improvements observed with CACE-NET. Another dataset that showcases significant benefits is Aarhus, particularly layer 3. Here, the model improves AUC by 6.21% and Accuracy by 10.84%. Likewise, in the Kapferer dataset, layer 2 exhibits substantial gains of 8.19% in AUC and 10.55% in Accuracy. These findings suggest that even in moderately structured networks, cross-layer coupling serves as a valuable supplement to enhance predictive power.

Notably, CACE-NET does not always outperform all baselines on every layer. For example, Aarhus layer 2 and Enron layer 2 exhibit only limited improvement or even degradation, indicating that cross-layer information does not always yield net benefits. For Aarhus layer 2—the densest layer among the five—it already contains abundant intra-layer information. While cross-layer coupling signals are beneficial for assisting relatively sparser layers, they provide limited additional value for this dense layer itself. Moreover, when structural disparity is high, adversarial training may suppress some useful features, allowing noise to have a larger impact, which explains why our model's performance is sometimes surpassed by certain baselines on this layer. For Enron layer 2, its structural properties also differ markedly compared to layer 1. Although it has fewer nodes than

Table 2: Ablation study on real-world datasets in terms of AUC and Accuracy(Acc).

(a) AUC

| Dataset | Layer | –L | –A | –F | CACE-NET |
|---|---|---|---|---|---|
| Aarhus | 1 | 0.9073 | 0.6271 | 0.9213 | **0.9851** |
| | 2 | 0.5678 | 0.6475 | 0.6188 | **0.7861** |
| | 3 | 0.9241 | 0.6419 | 0.9496 | **0.9853** |
| | 4 | 0.8623 | 0.7137 | 0.8795 | **0.9612** |
| | 5 | 0.8857 | 0.7708 | 0.9308 | **0.9627** |
| Enron | 1 | 0.9295 | 0.9195 | 0.9431 | **0.9905** |
| | 2 | 0.8601 | 0.7082 | 0.9070 | **0.9630** |
| Kapferer | 1 | 0.8708 | 0.7146 | 0.8800 | **0.9110** |
| | 2 | 0.7409 | 0.7045 | 0.7981 | **0.8503** |
| | 3 | 0.8635 | 0.7254 | 0.8604 | **0.9222** |
| | 4 | 0.8690 | 0.7288 | 0.6980 | **0.9462** |
| London | 1 | 0.8446 | 0.6388 | 0.8299 | **0.9158** |
| | 2 | 0.8072 | 0.5879 | 0.7495 | **0.8845** |
| | 3 | 0.8749 | 0.5926 | 0.8745 | **0.9025** |
| Reddit | 1 | 0.8900 | 0.8834 | 0.9007 | **0.9067** |
| | 2 | 0.8675 | 0.8566 | 0.8772 | **0.8976** |

(b) Acc

| Dataset | Layer | –L | –A | –F | CACE-NET |
|---|---|---|---|---|---|
| Aarhus | 1 | 0.8349 | 0.5175 | 0.8683 | **0.9460** |
| | 2 | 0.5067 | 0.5400 | 0.5133 | **0.7267** |
| | 3 | 0.9107 | 0.6339 | 0.8839 | **0.9286** |
| | 4 | 0.7638 | 0.6558 | 0.8141 | **0.9397** |
| | 5 | 0.8016 | 0.7127 | 0.8587 | **0.9111** |
| Enron | 1 | 0.8623 | 0.8656 | 0.8686 | **0.9529** |
| | 2 | 0.7765 | 0.6330 | 0.8283 | **0.8794** |
| Kapferer | 1 | 0.7863 | 0.6410 | 0.8077 | **0.8547** |
| | 2 | 0.7075 | 0.6226 | 0.6887 | **0.7594** |
| | 3 | 0.7788 | 0.6779 | 0.7548 | **0.8750** |
| | 4 | 0.7957 | 0.6304 | 0.6609 | **0.8739** |
| London | 1 | 0.7657 | 0.5571 | 0.7790 | **0.8707** |
| | 2 | 0.7688 | 0.5279 | 0.6506 | **0.9074** |
| | 3 | 0.7989 | 0.4788 | 0.8016 | **0.8624** |
| Reddit | 1 | 0.8222 | 0.8231 | 0.8299 | **0.8333** |
| | 2 | 0.7875 | 0.7931 | 0.8070 | **0.8213** |

layer 1, it has a higher average degree and greater density, indicating a denser network. At the same time, its heterogeneity index is substantially lower, suggesting a more homogeneous degree distribution with limited node-level variability. In other words, the structural patterns within layer 2 are relatively simple. A single-layer model can effectively capture the major connectivity rules, while cross-layer embeddings offer limited complementary value.

In summary, CACE-NET addresses key limitations of existing methods by effectively leveraging cross-layer coupling information. It delivers substantial improvements in prediction accuracy, particularly in settings with sparse intra-layer structure or strong interlayer dependencies, demonstrating its strength in extracting and utilizing complex multi-layer relationships.

### 5.3 ABLATION STUDY

To assess the contribution of each component in our proposed model, we performed an ablation study by constructing three variants: (i) CACE-NET (–L), which removes the layer-wise representation extractor; (ii) CACE-NET (–A), which omits the adversarial coupling representation encoder; and (iii) CACE-NET (–F), which replaces the self-attention fusion mechanism with simple averaging. Notably, in the –L and –F variants, the self-attention fusion becomes ineffective due to the absence of either intra-layer or cross-layer embeddings, so link prediction in these cases relies solely on the available representation.

Table 2 demonstrates that the full CACE-NET consistently outperforms all ablated variants, validating the necessity of each module. The layer-wise representation extractor and self-attention fusion are both critical: for example, in London layer 1, removing either causes AUC drops of 7.12% (–L) and 8.59% (–F), and Acc decreases of 10.5% and 9.17%, respectively. This underscores the importance of capturing intra-layer structure and adaptive fusion for effective link prediction.

The most pronounced performance degradation occurs when the adversarial coupling representation encoder is removed (–A), highlighting the significance of adversarially learned cross-layer information. In London layer 1, AUC and Acc decrease by 27.7% and 31.36%, when this module is omitted. This result emphasizes the critical role of inter-layer dependencies in multi-layer network link prediction and demonstrates the effectiveness of our approach in leveraging such information.

Notably, some network layers are less sensitive to the removal of cross-layer coupling. For example, in Aarhus layer 2, eliminating the coupling extractor causes smaller declines in AUC and Acc than other ablations. In this case, the layer already contains rich intra-layer structure, reducing its reliance on cross-layer information. However, such cases are rare; overall, the results strongly support the effectiveness and robustness of CACE-NET across a wide range of multi-layer network scenarios.

## 6 CONCLUSION

In this paper, we introduced CACE-NET, a novel framework for link prediction in multi-layer networks that incorporates cross-layer coupling relationships via adversarial training. By combining a layer-wise representation extractor, an adversarial coupling encoder, and an adaptive fusion predictor, our approach effectively models both intra-layer structures and hidden inter-layer dependencies. Extensive experimental results show that CACE-NET consistently outperforms existing methods, offering improvements in both predictive accuracy and robustness. For future work, we aim to expand it to dynamic multi-layer networks and a wider range of practical applications.

## ETHICS STATEMENT

Our CACE-NET method improves link prediction in multi-layer networks, supporting applications in social network analysis, transportation systems, and infrastructure monitoring. While enhancing prediction accuracy can offer valuable insights, there is a potential risk of misuse, such as inferring sensitive relationships or personal behaviors. We strongly encourage practitioners to apply this technology responsibly, respecting privacy, data protection, and ethical guidelines. The primary goal of our work is to advance network modeling and scientific understanding, and we advocate for careful, responsible use of these methods in real-world scenarios.

## REPRODUCIBILITY STATEMENT

We have made efforts to ensure the reproducibility of our work. Detailed experimental settings are provided in Appendix D.4. Additionally, the source code for CACE-NET is publicly available in an anonymous repository at `https://anonymous.4open.science/r/CACE-Net-B110/`. These measures are intended to facilitate the verification and replication of our results by other researchers in the field.

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

# A BACKGROUND

## A.1 GRAPH CONVOLUTIONAL NETWORK (GCN)

GCN by Kipf & Welling (2017) is a neural network architecture that generalizes convolution operations to non-Euclidean domains such as graphs. Given a target layer graph $G_t = (V_t, E_t)$ with an adjacency matrix $A_t \in \mathbb{R}^{|V_t| \times |V_t|}$ and a node feature matrix $X_t \in \mathbb{R}^{|V_t| \times d}$, we first compute the self-loop enhanced adjacency matrix as $\tilde{A}_t = A_t + I$, where $I$ is the identity matrix. The corresponding degree matrix is denoted as $\tilde{D}_t$ with entries $\tilde{D}_{ii} = \sum_j \tilde{A}_{ij}$. The node representation at layer $l + 1$ is updated using the standard GCN propagation rule:

$$H_t^{(l+1)} = \sigma \left( \tilde{D}_t^{-\frac{1}{2}} \tilde{A}_t \tilde{D}_t^{-\frac{1}{2}} H_t^{(l)} W_t^{(l)} \right), \tag{17}$$

where $H_t^{(l)}$ is the hidden representation of nodes at the $l$-th layer, $W_t^{(l)}$ is the trainable weight matrix, and $\sigma(\cdot)$ is a non-linear activation function, typically chosen as ReLU.

## A.2 FREELB

To enhance the robustness and generalization ability of our multi-layer network representation framework, we integrate the FreeLB adversarial training strategy (Zhu et al., 2020). This approach introduces adversarial perturbations in the continuous embedding space, and optimizes the model by accumulating parameter gradients across multiple adversarial ascent steps.

Specifically, let us denote the input embedding representation of node pairs as $H \in \mathbb{R}^{n \times d}$ and the corresponding training labels as $y$. The adversarial training in FreeLB can be formulated as solving the following minimax optimization problem:

$$\min_\theta \mathbb{E}_{(H,y) \sim \mathcal{D}} \left[ \max_{\|\delta\|_\infty \leq \varepsilon} \mathcal{L}(f_\theta(H + \delta), y) \right], \tag{18}$$

where $\delta$ represents adversarial perturbations bounded by an $\ell_\infty$ ball $\|\delta\|_\infty \leq \varepsilon$, $\theta$ denotes the parameters of the model, and $\mathcal{L}$ is a suitable loss function such as binary cross-entropy. The inner maximization is approximately solved via $T$ iterative steps of projected gradient ascent (PGD):

$$\delta_{t+1} = \Pi_{\|\delta\|_\infty \leq \varepsilon} \left( \delta_t + \alpha \cdot \nabla_\delta \mathcal{L}(f_\theta(H + \delta_t), y) \right), \tag{19}$$

where $\alpha$ denotes the perturbation step size, During the adversarial training procedure, FreeLB further accumulates parameter gradients across these multiple perturbation steps, updating the model parameters once per training iteration as follows:

$$\nabla_\theta \mathcal{L}_{\text{FreeLB}} = \frac{1}{T} \sum_{t=1}^{T} \nabla_\theta \mathcal{L}(f_\theta(H + \delta_t), y). \tag{20}$$

By leveraging this accumulated gradient, FreeLB effectively creates a richer set of adversarial embedding examples around each training sample, thereby enhancing the model's local robustness and consistency. Unlike conventional single-step adversarial training, FreeLB provides superior generalization capabilities without significantly increasing computational complexity.

## A.3 COUPLING IN MULTI-LAYER NETWORKS

A multi-layer network can be understood as a stack of $K$ graphs $G = \{G_1, \ldots, G_K\}$, each graph (or layer) encoding one specific relation among the same set of entities $V$. Taking a social scenario as an example, the layer $G_{\text{friend}}$ captures friendship ties, $G_{\text{work}}$ models professional collaboration, while $G_{\text{chat}}$ records instant–message interactions. Although every layer is a complete graph in its own right, information is coupled across layers because all layers refer to the same entities. To make this coupling explicit we treat an entity $u \in V$ as a collection of replica nodes $u_1, u_2, \ldots, u_K$ one per layer. For modeling purposes, we write $\pi(u_k) = u$ to indicate that all replicas share the same physical identity. This many-to-one mapping is the sole conduit through which information can flow from one layer to another.

---

**Algorithm 1:** CACE-NET Workflow

**Input:** A multi-layer network graph $G$.

**Output:** Estimate the probability of a link between each pair of nodes that have not observed a link

**1 for** *Treat each layer as the target layer in turn* **do**

**2**    **for** *each epoch* **do**

**3**       Obtain $H_{(u_i,v_i)}$ by Equation2;

**4**       Obtain $\widetilde{H}_{(u,v)}$ by Equation4;

**5**       Obtain $\overline{H}_{(u_t,v_t)}$ by Equation11;

**6**       Calculate $\mathcal{L}_D$, $\mathcal{L}_{LP}$ and $\mathcal{L}$ by Equation7, Equation13 and Equation14;

**7**       Initialize $\delta^{(0)} \leftarrow 0$;

**8**       **for** $t \leftarrow 1$ **to** $T$ **do**

**9**          Compute gradient: $g \leftarrow \nabla_{\delta^{(t-1)}} \mathcal{L}_D(f_\theta(h + \delta^{(t-1)}))$;

**10**          $\delta^{(t)} \leftarrow \Pi_{\|\delta\| \leq \epsilon} \left[ \delta^{(t-1)} + \alpha \cdot \text{sign}(g) \right]$;

**11**       **end**

**12**       Inject $(h + \delta^{(T)})$ into $f_\theta$ to obtain robust representation;

**13**       Perform backpropagation of $\mathcal{L}$ and update all parameters (generator $f_\theta$, discriminator $D_\phi$, etc.);

**14**    **end**

**15**    **for** *each unobserved link* $(u_t, v_t)$ *in the target layer* $G_t$ **do**

**16**       Calculate $P_{(u_t,v_t)}$ by Equation12;

**17**    **end**

**18 end**

---

In the context of link prediction, one of the layers is selected as the target layer , where we aim to predict missing links between nodes. The target layer contains the actual relationships we wish to predict. The auxiliary layers consist of the remaining layers in the multi-layer network. These auxiliary layers serve as complementary sources of information for the target layer, providing additional structural and relational context that can help improve the prediction accuracy. While the target layer contains the target relationships, the auxiliary layers offer broader context that aids in predicting those relationships. These layers are coupled with the target layer, meaning that changes or relationships in one layer can influence the predictions made in the target layer. The coupling effect captures how the structure and relationships in the auxiliary layers can influence the link prediction task in the target layer. This coupling enables information to flow across layers, improving the robustness and accuracy of predictions by incorporating both local and cross-layer dependencies.

## B   WORKFLOW OF CACE-NET

The workflow of CACE-NET, as detailed in Algorithm 1, proceeds as follows: (1) For each layer $G_t$ considered as the target layer, repeat the following steps. (2) For each training epoch: (a) Compute intra-layer pair embeddings $H_{(u_i,v_i)}$ using Equation 2; (b) Generate cross-layer embeddings $\widetilde{H}_{(u,v)}$ via Equation 4; (c) Fuse these to obtain $\overline{H}_{(u_t,v_t)}$ according to Equation 11; (d) Calculate the discriminator loss $\mathcal{L}_D$, link-prediction loss $\mathcal{L}_{LP}$, and total loss $\mathcal{L}$ using Equation 7, Equation 13, and Equation 14; (e) Initialize the adversarial perturbation $\delta^{(0)} \leftarrow 0$, and for $t = 1, \ldots, T$, update $\delta^{(t)} = \Pi_{\|\delta\|_\infty \leq \epsilon}\left( \delta^{(t-1)} + \alpha \, \text{sign}(\nabla_{\delta^{(t-1)}} \mathcal{L}_D(f_\theta(H + \delta^{(t-1)}))) \right)$; (f) Inject $(H + \delta^{(T)})$ into $f_\theta$ to obtain robust representations, and backpropagate $\mathcal{L}$ to update all parameters (including generator $f_\theta$ and discriminator $D_\phi$). (3) After training, for each unobserved link $(u_t, v_t)$ in $G_t$, compute the link probability $P_{(u_t,v_t)}$ using Equation 12.

# C  THEORETICAL ANALYSIS

## C.1  STABILITY OF ADVERSARIAL TRAINING (PROPOSITION 1)

In CACE-NET, the adversarial coupling representation encoder is trained via a min-max optimization detailed in Equation 8: $\min_\theta \max_\phi \mathbb{E}_{(u,v)\sim E_{train}} \sum_{i=1}^{K} \mathbb{1}[y_{(u,v)} = i] \log D_\phi(\widetilde{H}_{(u,v)})_i$, where $f_\theta$ denotes the generator producing cross-layer embeddings $\widetilde{H}_{(u,v)}$ for node pair $(u, v)$, and $D_\phi$ denotes the discriminator predicting the layer of origin. The indicator function $\mathbb{1}[y_{(u,v)} = i]$ equals 1 if the true layer label of $(u, v)$ is $i$, and 0 otherwise.

**Setup.** Let $\widetilde{H}_{(u,v)} = f_\theta(H_{(u_i,v_i)} + \delta)$ and we define $\mathcal{L}_D(H; \theta, \phi, y) := -\log D_\phi(f_\theta(H))_y$.

**Assumptions.** Some of the key assumptions are as follows:
**A1:** $f_\theta$ and $D_\phi$ are $L_f$- and $L_D$-Lipschitz in their inputs.
**A2:** $\mathcal{L}_D$ is $L$-smooth with bounded gradients.
**A3:** Learning rates are sufficiently small.
**A4:** FreeLB uses $T$ PGD steps with step size $\alpha$ and budget $\epsilon$ (Appendix A.2).

**Proposition 1 (Robust coupling under bounded perturbations).** Under (**A1**)–(**A4**), the alternating updates for $\theta$ and $\phi$ produce a bounded parameter sequence, and the induced embedding sequence $\{\widetilde{H}_{(u,v)}\}$ is stable (Cauchy up to stochastic noise).

*Proof sketch.* FreeLB enforces $\|\delta\|_\infty \leq \epsilon$. Lipschitz continuity yields $\|f_\theta(H + \delta) - f_\theta(H)\| \leq L_f\epsilon$ and bounded changes in $\mathcal{L}_D$. With bounded gradients and small steps, parameter updates are bounded. Smoothness gives a local contraction in expectation for the alternating game near stationary points, implying stability of $\widetilde{H}_{(u,v)}$. $\qquad\square$

**Embedding stability bound.** Writing $\widetilde{H}_{(u,v)}^t = f_{\theta^t}(H + \delta^t)$, we have

$$\|\widetilde{H}_{(u,v)}^{t+1} - \widetilde{H}_{(u,v)}^t\| \leq L_f \|\theta^{t+1} - \theta^t\| + L_f \|\delta^{t+1} - \delta^t\| \leq c_\theta\, \eta + c_\delta\, \alpha, \tag{21}$$

for constants $c_\theta, c_\delta > 0$ depending on local Lipschitz and gradient bounds.

## C.2  LIPSCHITZ PROPERTY OF FUSION (LEMMA 1)

To control how perturbations propagate through the fusion module, we establish that the multi-head attention-based fusion is a Lipschitz mapping under mild operator-norm constraints on its projection matrices and the usual $1/\sqrt{d_k}$ scaling. This guarantees that small changes in its inputs $[\,H_{(u_t,v_t)} \parallel \widetilde{H}_{(u,v)}\,]$ (e.g., from adversarial perturbations or stochastic optimization) are not amplified by fusion.

**Lemma 1 (Lipschitz fusion).** Suppose $\|W_i^Q\|, \|W_i^K\|, \|W_i^V\| \leq c$ and attention uses scaling by $1/\sqrt{d_k}$. Then the multi-head attention mapping $x \mapsto \overline{H}(x)$ is $C(c, h)$-Lipschitz.

*Proof sketch.* Each affine map is $c$-Lipschitz. Softmax is 1-Lipschitz on bounded domains after scaling; multiplication by $V_i$ preserves Lipschitzness up to $c$. Concatenating $h$ heads scales the constant by $h$. Hence $\overline{H}$ is Lipschitz with $C = O(hc^2)$. $\qquad\square$

## C.3  END-TO-END PREDICTION STABILITY

We bound the sensitivity of the entire prediction pipeline to ensure that robustness in learned representations translates to stable final outputs. Specifically, combining Proposition 1 (embedding stability), Lemma 1 (Lipschitz fusion), and the Lipschitz property of the classifier shows that small input perturbations in $[\,H_{(u_t,v_t)} \parallel \widetilde{H}_{(u,v)}\,]$ result in proportionally small changes in predicted probabilities. Across training epochs, prediction changes are controlled by the learning rate and adversarial step size, guaranteeing stable optimization.

**Lemma 2 (Prediction stability).** Let $\sigma$ and the MLP $f$ in Equation 12 be $L_\sigma$- and $L_f$-Lipschitz, respectively. Under Proposition 1 and Lemma 1, there exists $K > 0$ such that for any two inputs $x = [H_{(u_t,v_t)} \parallel \widetilde{H}_{(u,v)}]$ and $x'$,

$$\left| P_{(u_t,v_t)}(x) - P_{(u_t,v_t)}(x') \right| \leq K \|x - x'\|. \tag{22}$$

In particular, across successive epochs,

$$\left| P_{(u_t, v_t)}^{t+1} - P_{(u_t, v_t)}^t \right| \leq K \left( c_\theta\, \eta + c_\delta\, \alpha \right), \tag{23}$$

so predictions are stable for sufficiently small learning rates $\eta$ and adversarial step sizes $\alpha$.

*Proof sketch.* Compose the Lipschitz constants from fusion (Lemma 1) and the classifier ($w^T \circ f \circ \sigma$). Stability of $x$ over epochs follows from the embedding bound and the fact that $H_{(u_t, v_t)}$ varies smoothly with parameters by standard GCN Lipschitz arguments (Appendix A.1). $\qquad\square$

Collectively, our results show that: (i) the adversarial coupling encoder is stable under bounded perturbations (Proposition 1), yielding bounded parameter and embedding sequences; (ii) the attention-based fusion module is a Lipschitz mapping (Lemma 1), so it does not amplify small perturbations in $[H_{(u_t, v_t)} \parallel \widetilde{H}_{(u,v)}]$; and (iii) the entire prediction pipeline is stable (Lemma 2), with sensitivity bounded by a constant $K$ and epoch-to-epoch drift scaling with the learning rate $\eta$ and adversarial step size $\alpha$. These guarantees formally justify the robustness of CACE-NET and motivate practical design choices (e.g., modest $\alpha$, appropriate $T$, and bounded operator norms for attention projections) to promote stable training and reliable link predictions.

### C.4 Complexity Analysis

Both time and memory costs of CACE-NET are well controlled. The model consists of three main components: a layer-wise representation extractor, an adversarial coupling encoder, and an adaptive fusion link predictor.

Let $|Train|$ and $|Test|$ denote the number of training and testing link samples, respectively. (i) The layer-wise extractor (GCN) has time complexity $O(|E| + |V| \cdot dim_n)$, where $|E|$ is the total number of edges, $|V|$ is the total number of nodes, and $dim_n$ is the node feature dimension. (ii) The adversarial encoder (generator, discriminator, and FreeLB) has complexity $O(|Train| \cdot dim)$, where $dim$ is the edge feature dimension. (iii) The adaptive fusion predictor uses attention ($O(|Train| \cdot dim^2)$) and a fully connected layer ($O(|Train| \cdot dim)$).

Overall, training complexity is $O(|E| + |V| \cdot dim_n + |Train| \cdot dim^2)$. During testing (without FreeLB), it is $O(|E_k| + |V| \cdot dim_n + |Test| \cdot dim^2)$, where $|E_k|$ is the number of edges in the target layer.

Regarding memory/space complexity, we store each layer as a sparse COO adjacency matrix and process samples in batches. This design ensures that CACE-NET is parameter-efficient and maintains low memory consumption.

## D  Further details on Experimental Settings

### D.1  Datasets

To evaluate the performance of CACE-NET fairly, we use several real-world multi-layer networks, which are described as follows:

(i) Aarhus Magnani et al. (2013): This network represents five distinct types of interactions among employees at the Aarhus Department of Computer Science. The layers correspond to interactions on Facebook, in leisure activities, at work, during co-authoring, and at lunch.

(ii) Enron Tang et al. (2012): This dataset captures the interactions between Enron employees, with two layers that represent the relationships with their superiors and colleagues, respectively.

(iii) Kapferer De Domenico et al. (2014): This 4-layer network was observed over a ten-month period in a tailor shop, capturing various aspects of relationships, such as work, assistance, friendship, and emotional ties.

(iv) London De Domenico et al. (2014): This dataset represents the railway stations in London, with three layers representing connections via the underground, overground, and the Docklands Light Railway (DLR).

(v) Reddit Kumar et al. (2018): A 2-layer network based on Reddit posts, with hyperlinks between subreddits. The first layer captures hyperlinks in post titles, while the second captures hyperlinks in post bodies, representing different types of interactions within subreddits.

Table 3: Statistics of several multi-layer network datasets.

| Dataset | $|V|$ | $|E|$ | layer $k$ | $|V_k|$ | $|E_k|$ | $c_k$ | $\langle\mu_k\rangle$ | $H_k$ | $\rho_k$ |
|---|---|---|---|---|---|---|---|---|---|
| Aarhus | 61 | 620 | 1 | 60 | 193 | 0.6733 | 6.4333 | 1.2129 | 0.1090 |
| | | | 2 | 32 | 124 | 0.5404 | 7.7500 | 1.2268 | 0.2500 |
| | | | 3 | 25 | 21 | 0.2680 | 1.6800 | 1.3889 | 0.0700 |
| | | | 4 | 47 | 88 | 0.3925 | 3.7447 | 1.5143 | 0.0814 |
| | | | 5 | 60 | 194 | 0.6396 | 6.4667 | 1.6652 | 0.1096 |
| Enron | 151 | 261 | 1 | 142 | 133 | 0.0000 | 1.8732 | 2.6652 | 0.0133 |
| | | | 2 | 117 | 128 | 0.0000 | 2.1880 | 1.4925 | 0.0189 |
| Kapferer | 39 | 552 | 1 | 39 | 158 | 0.4580 | 8.1026 | 1.3467 | 0.2132 |
| | | | 2 | 39 | 223 | 0.4977 | 11.4359 | 1.2262 | 0.3009 |
| | | | 3 | 35 | 76 | 0.3101 | 4.3429 | 1.5058 | 0.1277 |
| | | | 4 | 37 | 95 | 0.3351 | 5.1351 | 1.5558 | 0.1426 |
| London | 369 | 441 | 1 | 271 | 312 | 0.0311 | 2.3026 | 1.1804 | 0.0085 |
| | | | 2 | 83 | 83 | 0.0000 | 2.0000 | 1.0723 | 0.0244 |
| | | | 3 | 45 | 46 | 0.0185 | 2.0444 | 1.0952 | 0.0465 |
| Reddit | 67,180 | 858,488 | 1 | 54,075 | 571,927 | 0.1844 | 21.1531 | 45.6860 | 0.0002 |
| | | | 2 | 35,776 | 286,561 | 0.1809 | 16.0197 | 24.3141 | 0.0002 |

The specific statistics of these multi-layer network datasets are summarized in Table 3. In the table, $c_k$ denotes the clustering coefficient for the $k$-th layer, $\langle\mu_k\rangle$ represents the average degree of nodes, and $H_k = \frac{\langle\mu_k^2\rangle}{\langle\mu_k\rangle^2}$ indicates the degree of heterogeneity Lü & Zhou (2011). Additionally, $\rho_k = \frac{2|E_k|}{|V_k|(|V_k|-1)}$ represents the density of each layer.

### D.2 BASELINES

To assess the performance of our proposed CACE-NET, we compare it with several state-of-the-art methods. These include traditional single-layer network approaches applied directly to the target layer, as well as specialized baseline methods designed for multi-layer networks:

**(i) CN (Kossinets, 2006):** This method counts the number of common neighbors between two nodes, under the assumption that nodes sharing more common neighbors are more likely to form a link.

**(ii) Jaccard (Jaccard, 1901):** This measure calculates the ratio of shared neighbors to the total number of neighbors across both nodes, making it particularly effective for networks with nodes having highly variable degrees.

**(iii) NSILR (Yao et al., 2017):** This approach proposes a node similarity index that leverages the relevance between layers in multi-layer networks, using both intra-layer and inter-layer representations.

**(iv) SEAL (Zhang & Chen, 2018):** SEAL first constructs adjacency subgraphs and then applies DGCNN to learn representations from these subgraphs. For comparison, we treat the multi-layer network as a single-layer network, using layer-specific information as attributes.

**(v) MultiSup (Shan et al., 2020):** This method introduces two novel metrics, friendship between neighbors (FoN) and friendship among auxiliary layers (Fal), to extract rich structural representations from all layers for link prediction.

**(vi) MADM (Luo et al., 2021):** MADM treats the integration of multi-layer information as a multi-attribute decision-making problem, calculating intra-layer similarities through resource allocation metrics and inter-layer similarities using cosine similarity.

**(vii) MNERLP (Mishra et al., 2022):** This method calculates node and edge relevance by utilizing both local and global representations, combining these factors to improve link prediction accuracy.

**(viii) HOPLP (Mishra et al., 2023):** HOPLP transforms the multi-layer network into a weighted single-layer network, considering the relative density of each layer, and iteratively calculates link likelihoods by accounting for longer paths between nodes.

**(ix) LUSTER (Yang et al., 2025):** LUSTER captures the shared latent space representation across multiple layers through adversarial training and orthogonal fusion, enhancing complementary information among layers to improve link prediction performance.

### D.3    EVALUATION METRICS

**(i) Accuracy (Acc).** Accuracy reflects the proportion of correctly predicted instances out of the total number of instances. It serves as a simple and direct measure of overall prediction correctness. The formula for Accuracy is given by:

$$\text{Accuracy} = \frac{\text{TP} + \text{TN}}{\text{TP} + \text{TN} + \text{FP} + \text{FN}} \quad (24)$$

where TP (True Positive) represents the count of correctly predicted positive samples, TN (True Negative) is the count of correctly predicted negative samples, FP (False Positive) refers to the incorrectly predicted positive samples, and FN (False Negative) represents the incorrectly predicted negative samples.

**(ii) Area Under the ROC Curve (AUC).** The AUC quantifies the area beneath the Receiver Operating Characteristic (ROC) curve, which visually depicts the trade-off between the True Positive Rate (TPR) and the False Positive Rate (FPR) at various decision thresholds. The True Positive Rate (TPR), also known as recall, is the fraction of actual positive instances that are correctly identified as positive:

$$\text{TPR} = \frac{\text{TP}}{\text{TP} + \text{FN}} \quad (25)$$

The False Positive Rate (FPR) represents the proportion of negative instances that are incorrectly classified as positive:

$$\text{FPR} = \frac{\text{FP}}{\text{FP} + \text{TN}} \quad (26)$$

The AUC value ranges from 0 to 1. A value of 1 signifies perfect classification, while 0.5 indicates random guessing. A higher AUC indicates superior performance in distinguishing between positive and negative samples.

In summary, both Accuracy and AUC provide important insights into the performance of the model. Higher values for both metrics suggest better model performance in link prediction tasks across a variety of real-world datasets.

### D.4    EXPERIMENTAL SETUP

All experiments were conducted on real-world multi-layer network datasets, which are typically sparse and imbalanced. To address this, for each positive link $(u, v)$, we sample a negative link $(u, v')$ by randomly choosing $v'$ from nodes not connected to $u$, maintaining a 1:1 ratio of positive to negative samples. Each dataset was split into training, validation, and test sets in an 8:1:1 ratio. Node features were initialized as identity matrices. GCN node embeddings had dimension 16, and edge features were formed by concatenating node embeddings (32 dimensions). Models were trained with Adam (Kingma & Ba, 2014) with initial learning rate 0.001, batch size 128, up to 1000 epochs, with early stopping based on validation. In the adaptive fusion link predictor, the number of heads in the multi-head attention mechanism was set to 2. For the adversarial perturbation, the hyperparameters $T$, $\alpha$, and $\epsilon$ were set to 3, 0.005, and 0.01, respectively. The trade-off parameter $\lambda$ in Equation14 was fixed to 0.5. All code was implemented in Python 3.11 with PyTorch 2.0 and run on an NVIDIA RTX 4090 GPU. Baseline methods were reproduced using recommended settings and further fine-tuned if needed for fair comparison.

## E    ADDITIONAL EXPERIMENTAL RESULTS

### E.1    GENERALIZATION ANALYSIS

To assess the generalization capability of CACE-NET under unseen distributions, we conducted a structural analysis between training layers and target layers across all datasets. Specifically, we quantified each layer's size ($|V_k|$, $|E_k|$), clustering coefficient ($c_k$), average degree ($\mu_k$), degree heterogeneity ($H_k$), and density ($\rho_k$), as summarized in Table 4 (in the Appendix). These metrics serve as proxies for measuring structural and statistical distributional differences between layers.

As shown in the results, CACE-NET demonstrates strong generalization even when the target layer differs significantly from the training layers. For instance, in the London dataset, the number of

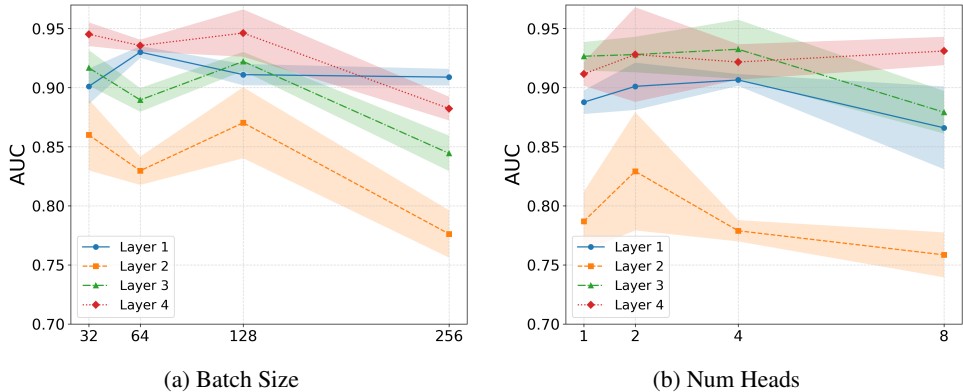

(a) Batch Size          (b) Num Heads

Figure 3: Effect of batch size and number of attention heads on the performance of CACE-NET on the Kapferer dataset.

Table 4: Effect of Adversarial Loss Weight $\lambda$ (AUC / Acc) on the Kapferer Dataset.

| $\lambda$ | 0.3 | 0.5 | 0.7 | 0.8 |
|---|---|---|---|---|
| Kapferer 1 | 0.8889 / 0.7949 | 0.9310 / 0.8077 | 0.8888 / 0.8120 | 0.8738 / 0.8205 |
| Kapferer 2 | 0.8337 / 0.7453 | 0.8264 / 0.7358 | 0.7800 / 0.7123 | 0.8438 / 0.7972 |
| Kapferer 3 | 0.9109 / 0.8029 | 0.8879 / 0.8029 | 0.8482 / 0.8029 | 0.9167 / 0.8462 |
| Kapferer 4 | 0.8546 / 0.7739 | 0.8983 / 0.8174 | 0.8633 / 0.8000 | 0.9071 / 0.8130 |

nodes varies from 45 to 271 across layers, and density changes from 0.0085 to 0.0465, yet CACE-NET achieves consistently high AUC scores. Similarly, in the Enron dataset, despite shifts in average degree and heterogeneity, the model maintains high AUCs of 0.9905 and 0.9630. These results highlight the robustness of CACE-NET to structural variations and distribution shifts across layers.

On the other hand, we also observe that in extreme cases of sparsity or size imbalance (e.g., Aarhus Layer 2, with a relatively high density of 0.25), the performance slightly drops. This may be attributed to the mismatch between the target layer's compact, densely connected structure and the sparser training layers, which could limit the model's ability to transfer learned representations. Such cases indicate that CACE-NET's generalization may degrade when the target distribution deviates significantly from the training distribution along multiple structural dimensions.

Overall, the empirical evidence suggests that CACE-NET is capable of generalizing to unseen distributions, especially when the underlying connectivity patterns are preserved, even if the target layer differs in scale or density.

## E.2 PARAMETER ANALYSIS

In this subsection, we investigate how key hyperparameters affect model performance, aiming to determine optimal settings. We focus on several important parameters, including `batch_size`, `num_heads`, as well as the adversarial loss weight $\lambda$ and adversarial perturbation settings $(T, \alpha, \epsilon)$, and conduct controlled experiments on the Kapferer dataset to assess their effects.

**Effect of Batch Size.** The choice of `batch_size` plays a crucial role in balancing training speed and generalization. As shown in Fig. 3a, increasing `batch_size` beyond a certain point leads to diminished generalization, with the best results achieved at a batch size of 128.

**Effect of Number of Attention Heads.** We also examine the effect of varying `num_heads`, which controls the number of attention heads and thus the model's ability to capture diverse information. Fig. 3b shows that for layers 0, 2, and 3, performance remains relatively stable across different values, with both 2 and 4 heads yielding competitive results. In contrast, layer 1 is more sensitive to this parameter, with two attention heads providing the best performance. Overall, setting `num_heads` to 2 offers a good trade-off and is selected as the default configuration.

Table 5: Performance (AUC / Acc) under different hyperparameter settings $(T, \alpha, \epsilon)$ across four layers on the Kapferer dataset.

| $(T, \alpha, \epsilon)$ | Layer 1 | Layer 2 | Layer 3 | Layer 4 |
|---|---|---|---|---|
| (3, 0.005, 0.01) | 0.9296 / 0.8504 | 0.8549 / 0.7594 | 0.9197 / 0.8558 | 0.9010 / 0.8261 |
| (3, 0.003, 0.01) | 0.9081 / 0.8333 | 0.8370 / 0.7511 | 0.8739 / 0.8240 | 0.9185 / 0.8609 |
| (3, 0.01, 0.03) | 0.8670 / 0.7821 | 0.8371 / 0.7453 | 0.8793 / 0.8077 | 0.8683 / 0.7478 |
| (5, 0.006, 0.03) | 0.9011 / 0.7863 | 0.8199 / 0.7642 | 0.9043 / 0.7115 | 0.8647 / 0.7391 |
| (5, 0.002, 0.01) | 0.8836 / 0.7949 | 0.7944 / 0.7123 | 0.8645 / 0.7933 | 0.9044 / 0.7957 |

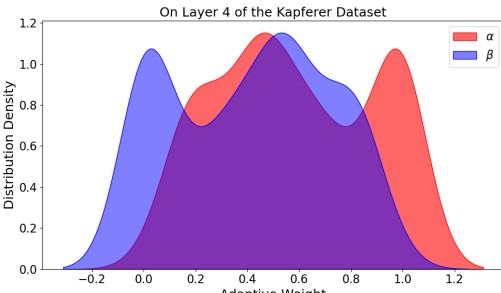

Figure 4: Distribution of fusion attention weights.

**Effect of Adversarial Loss Weight** $\lambda$ We examined the impact of varying the adversarial loss weight $\lambda$ on the Kapferer dataset while fixing other hyperparameters. As shown in Table 4, different layers respond to $\lambda$ in distinct ways: larger values (e.g., $\lambda = 0.7$ or $0.8$) sometimes benefit heterogeneous layers such as Layer 3 and 4, but may also lead to unstable results, particularly in Layer 2. In contrast, smaller values (e.g., $\lambda = 0.3$) yield competitive results in certain cases but underperform overall. The setting $\lambda = 0.5$ achieves the most balanced trade-off, delivering strong performance in Layer 1 and stable results across other layers. Based on this balance between accuracy and robustness, we adopt $\lambda = 0.5$ as the default configuration in subsequent experiments.

**Effect of Adversarial Perturbation Parameters.** To examine the sensitivity of CACE-NET to adversarial perturbation settings, we conducted experiments on the Kapferer dataset with varying combinations of $(T, \alpha, \epsilon)$. The results are reported in Table 5, which presents AUC and ACC performance across the four layers. As shown in the table, moderate perturbations with a small budget consistently achieve comparatively better performance, enhancing both AUC and ACC across most layers. In contrast, larger perturbation budgets tend to over-regularize the model, leading to performance degradation in one or more layers. Based on these findings, we selected $(T, \alpha, \epsilon) = (3, 0.005, 0.01)$ for all subsequent experiments, as it strikes a balance between effective regularization and the preservation of useful information.

### E.3 CASE STUDY

To gain deeper insights into how CACE-NET leverages intra-layer and cross-layer coupling information for link prediction, we conduct the case study on the Kapferer dataset. Specifically, we examine the distribution of fusion attention weights and the structural coupling between node pairs across layers. The analysis in the following subsubsections illustrates how our model dynamically integrates complementary information from auxiliary layers to improve prediction accuracy.

### E.3.1 ATTENTION WEIGHT ANALYSIS

We selected the fourth layer of the Kapferer dataset for an in-depth case study to analyze the distribution of fusion attention weights, as depicted in Fig. 4. In this figure, $\alpha$ represents intra-layer attention weights, while $\beta$ denotes cross-layer coupling attention weights. The bimodal distribution observed in the attention curves indicates significant variability in the attention allocation to different node pairs within this layer. Specifically, cross-layer coupling embeddings primarily provide sup-

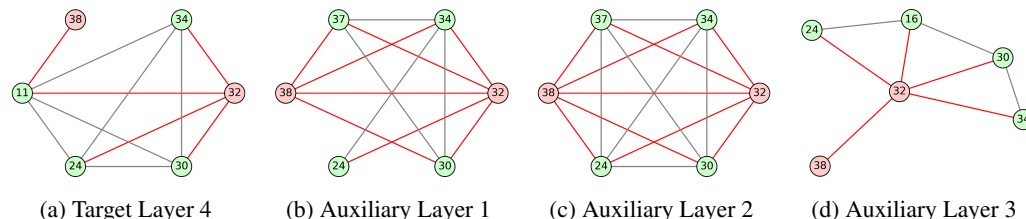

(a) Target Layer 4  (b) Auxiliary Layer 1  (c) Auxiliary Layer 2  (d) Auxiliary Layer 3

Figure 5: Schematic diagram of the local structure and cross-layer relationship of multi-layer networks.

plementary information and support. This observation highlights the flexibility of the self-attention mechanism in our model, effectively balancing the relative contributions of intra-layer and cross-layer coupling representations for the prediction task.

### E.3.2 STRUCTURAL COUPLING ANALYSIS

To further illustrate the advantages provided by auxiliary layers, we performed a structural analysis of node pairs exhibiting relatively high cross-layer coupling attention weights. As shown in Fig. 5, we highlight a representative node pair (32, 38), where cross-layer coupling information transfer markedly enhances prediction accuracy. This node pair features sparse connectivity in the target layer, which alone is insufficient for accurate predictions. However, across several auxiliary layers, this node pair shares numerous common neighbors, demonstrating robust structural coupling. By dynamically emphasizing cross-layer coupling embeddings, our model effectively integrates complementary structural information, thus achieving accurate predictions. This example underscores the significance of cross-layer coupling and demonstrates the efficacy of our fusion-based representation approach.

## F  LIMITATIONS

Despite the strong performance of our method, several limitations remain. First, our approach presumes that auxiliary layers are both structurally and semantically aligned with the target layer, an assumption that may not always be valid in real-world scenarios. Second, although the FreeLB-based adversarial training enhances robustness, it introduces sensitivity to the choice of perturbation magnitude and other hyperparameters, which can impact training stability. Third, the framework focuses on static graphs, with computational costs scaling with layer count, potentially limiting scalability. Future research could focus on developing more stable adversarial training techniques and extending the model to efficiently handle dynamic or large-scale multi-layer networks.

## G  USE OF LARGE LANGUAGE MODELS

We used Large Language Models (LLMs) to assist in polishing the manuscript. All content generated with the help of LLMs was carefully reviewed, verified, and edited by the authors to ensure accuracy and originality. We take full responsibility for all content in the paper, including any parts assisted by LLMs.

