# OpenReview forum: "CACE-Net: Cascade Coupling Effect for Link Prediction in Multi-layer Networks"
_ICLR.cc/2026/Conference — ICLR 2026 Conference Withdrawn Submission_

### Official Review · Reviewer_eAbx · 2025-10-30

**Soundness:** 2
**Presentation:** 3
**Contribution:** 2
**Rating:** 4
**Confidence:** 3

**Summary:**

This paper introduces CACE-NET, a novel framework for link prediction in multi-layer networks. The central thesis is that existing methods often fail to capture the "cascade coupling effect," a phenomenon where structural changes in one layer dynamically propagate to influence link formation in others. To address this gap, CACE-NET integrates three key components: a layer-wise representation extractor for modeling intra-layer topology, an adversarial coupling representation encoder to learn shared, layer-invariant embeddings that capture cross-layer dependencies, and an adaptive fusion predictor that uses an attention mechanism to dynamically weigh and combine these intra- and cross-layer signals. The primary contribution is the novel use of an adversarial training scheme to explicitly model and leverage latent, dynamic inter-layer dependencies, which is further supported by theoretical analysis.

**Strengths:**

1. The paper's framing of the problem is a significant strength. It moves beyond the conventional paradigm of simply "fusing cross-layer information" to deeply investigate the "cascade coupling effect," perceptively analyzing how structural changes propagate as a chain reaction across layers. This perspective is both novel and highly insightful.

2. The use of a generator-discriminator framework to learn embeddings that are invariant to their layer of origin is a particularly clever solution for distilling shared coupling information.

3. The contribution is significantly strengthened by the inclusion of theoretical analysis, providing formal proofs for the model's stability and robustness.

4. The commitment to reproducibility, demonstrated by the detailed experimental setup in the appendix and the promise of code release, is crucial for ensuring the transparency and verifiability of the work.

**Weaknesses:**

1. Scalability: The current design, with independent GCNs per layer and iterative adversarial training, may not scale efficiently to networks with a very large number of layers or nodes.

2. Dependence on Auxiliary Layer Quality: The framework's performance relies on informative auxiliary layers. It is unclear how the model would perform if these layers were noisy or semantically irrelevant.

3. Hyperparameter Sensitivity: The model introduces several hyperparameters for its adversarial and fusion modules. Their complex interactions could make tuning on new datasets difficult.

**Questions:**

1. What would be the effect of introducing a purely random graph as an auxiliary layer? Could the discriminator successfully identify and isolate the embeddings from this noise layer, thereby compelling the generator to ignore it, or would its presence lead to a significant degradation in overall performance?

2. The case study (Figure 4) on attention weight distribution is a good starting point. To make the qualitative analysis more concrete and compelling, the authors could walk through a specific link prediction instance. For example, identify a node pair (u, v) that receives a particularly high cross-layer attention weight (β), and then, using the schematics in Figure 5, pinpoint the exact structural information in a specific auxiliary layer (e.g., a critical shared neighbor).

3. The term "cascade effect" inherently implies a temporal component. A natural and potentially high-impact next step for this research would be to extend the framework to dynamic multi-layer networks, where the graph topology evolves over time. Have the authors considered how CACE-NET might be adapted for such a scenario?

---

### Official Review · Reviewer_5v61 · 2025-10-31

**Soundness:** 2
**Presentation:** 2
**Contribution:** 2
**Rating:** 2
**Confidence:** 4

**Summary:**

This paper proposes CACE-NET, a framework for link prediction in multi-layer networks that models cascade coupling effects. The model consists of three main components: (1) a layer-wise feature extractor using GCNs, (2) an adversarial coupling encoder that captures cross-layer dependencies with FreeLB-based training, and (3) an adaptive fusion link predictor that integrates intra- and inter-layer information through an attention mechanism. The authors report experiments on five real-world datasets (Aarhus, Enron, Kapferer, London, and Reddit), claiming consistent improvements over existing baselines.

**Strengths:**

1. This paper recognizes the importance of cross-layer dependencies, which is a real issue in multilayer networks.
2. The paper has a clear modular decomposition (per-layer encoders -> adversarial coupling -> attention fusion).

**Weaknesses:**

1. The introduction section lacks conceptual grounding in link prediction. The first paragraph focuses solely on multi-layer networks without defining link prediction or explaining its significance. The second paragraph immediately discusses the challenge posed by "cascade dynamics" towards link prediction. Given that link prediction is the central theme of this paper, its insufficient introduction weakens the motivation and logical flow.
2. Although the authors claim to include state-of-the-art (SOTA) baselines, key recent works are missing. For example, Ren et al. (2024) is cited in the introduction to represent the third category of existing methods (“Extended cross-layer embedding methods”), yet it is not included in the experimental comparison. Given that Ren et al. (2024) was published at a top-tier conference, this inconsistent omission undermines the fairness and completeness of the evaluation.
3. In the related work section, the authors state, “Recent studies demonstrate a certain degree of correlation between the topological features of different layers in multi-layer networks (Szell et al., 2010; Lee et al., 2015).” However, both cited works are more than a decade old, making this claim inconsistent with the term “recent” and suggesting insufficient engagement with up-to-date research.
4. Table 1 is unclear. It uses both bold and underlined numbers without any explanation of their meaning. The table also reports improvement percentages but does not describe how these percentages were calculated. From context, it appears the improvement refers to the percentage increase of the bold number over the underlined number, but this should be explicitly stated.
5. The reported “up to +13.29% AUC” result highlights only the best-case scenario. Median or average improvements across datasets and layers are not emphasized. A more balanced statistical summary (e.g., mean ± standard deviation) should be presented to strengthen the reliability of the reported results.

**Questions:**

Please see the weaknesses

---

### Official Review · Reviewer_6iUW · 2025-10-31

**Soundness:** 3
**Presentation:** 3
**Contribution:** 2
**Rating:** 4
**Confidence:** 3

**Summary:**

The paper introduces CACE-NET, a novel framework for link prediction in multi-layer networks.  CACE-NET addresses  the cascade coupling effects, where structural changes propagate across different network layers and influence link formation,  by incorporating three components: a Layer-wise Representation Extractor for intra-layer structure, an Adversarial Coupling Representation Encoder that uses adversarial training to learn robust, layer-invariant cross-layer dependencies, and an Adaptive Fusion Link Predictor that balances both types of information using an attention mechanism. Experimental results on real-world datasets show the effectiveness of CACE-NET.

**Strengths:**

- The adversarial coupling encoder leverages adversarial training (FreeLB) to learn  shared and layer-invariant representations

-  Uses an adaptive fusion link predictor with a multi-head attention mechanism to dynamically weigh and combine intra-layer structural information (local) and cross-layer coupling signals (global)

- Theoretical Stability Guarantees: Formal analysis establishes that the cross-layer embeddings are robust and stable under bounded adversarial perturbations (Proposition 1), and the fusion and prediction modules are Lipschitz continuous, ensuring controlled perturbation propagation and overall stability (Lemma 1 and Lemma 2)

- Provides substantial accuracy gains, particularly in layers characterized by sparse connectivity or low clustering, by compensating for missing intra-layer information using auxiliary layers

**Weaknesses:**

1. The approach assumes that auxiliary layers are both structurally and semantically aligned with the target layer, which may not always hold true in real-world scenarios

2. The FreeLB-based adversarial training introduces sensitivity to the choice of perturbation magnitude and other hyperparameters,  which can impact training stability

3. CACE-NET may show limited improvement or even degradation in layers that are already very dense, as cross-layer coupling provides limited complementary value when intra-layer structure is already abundant

4. Scalability concerns: Computational costs scale with the layer count, potentially limiting its scalability to very large multi-layer networks

5. CACE-NET's baseline comparison includes multilayer link prediction methods mostly employing adversarial training for shared latent space representation. However, there is an important body of studies in multilayer link prediction that CACE-NET does not explicitly compare against, particularly those methods exploiting multilayer GNN embeddings at node level as well as NN-learned node-pair structural features derived from overlapping neighborhoods.

**Questions:**

See above Weaknesses, and consider addressing them, perhaps giving higher priority to point 5.

---

### Official Review · Reviewer_F2xT · 2025-11-01

**Soundness:** 2
**Presentation:** 3
**Contribution:** 2
**Rating:** 6
**Confidence:** 3

**Summary:**

This paper aims to capture the cascade coupling effect in multi-layer networks, where variations in one layer can trigger a chain reaction across others. To model this phenomenon, the authors consider both intra-layer structural information and inter-layer interactions. The intra-layer structure is modeled through GNN-based aggregation, while the inter-layer dependencies are captured via a GAN-based mechanism, where the generator learns to align representations with the target layer and the discriminator distinguishes between target and auxiliary layers. Finally, the intra- and inter-layer embeddings are integrated through a multi-head attention module.

**Strengths:**

1. The paper addresses an important but underexplored problem: modeling both intra-layer and dynamic inter-layer coupling effects in multi-layer networks.

2. It provides theoretical and empirical analysis suggesting that the proposed framework can learn stable and robust cross-layer embeddings.

3. Experimental results demonstrate superior performance compared to several competitive baselines.

**Weaknesses:**

1. It would strengthen the paper to include a time complexity analysis, especially since the GAN-based adversarial training could increase computational overhead and reduce scalability.

2. The paper could also provide a sensitivity analysis of the hyperparameter $\lambda$, to better illustrate how it influences the balance between the loss terms and overall model performance.

**Questions:**

please refer to the weakness

---

### Official Review · Reviewer_ozsv · 2025-11-01

**Soundness:** 3
**Presentation:** 3
**Contribution:** 3
**Rating:** 2
**Confidence:** 3

**Summary:**

This paper proposes a new link prediction method to capture the cross-layer dependencies. CACE-NET uses a Layer-wise representation extractor that uses independent GCNs to model each layer's specific intra-layer topology, and use an adversarial coupling representation encoder to learn robust, layer-invariant cross-layer embeddings. Also, it use an adaptive fusion link predictor to integrate the intra-layer and cross-layer embeddings for the final prediction. Experiments on five real-world datasets show the effectiveness of proposed method.

**Strengths:**

1. This paper provides a rigorous theoretical foundation for deriving the proposed prediction stability.
2. The paper is easy to follow, with three main components.
3. Extensive experiments of this paper support the method proposed in this paper.

**Weaknesses:**

1. The paper's Related works and Baselines are outdated (except for LUSTER's). The authors should incorporate some of the latest research, such as CasMS[1].
2. The paper's own description of LUSTER states it "captures the shared latent space representation across multiple layers through adversarial training". This sounds functionally identical to CACE-NET's "Adversarial coupling representation encoder" which learns "shared, layer-invariant features". While CACE-NET outperforms LUSTER in Table 1, the didn't never explains why.
3. The paper is motivated by cross-layer cascade effect. However, the proposed model does not appear to model this sequential dependency. The adversarial encoder takes embeddings from all layers and learns a single embedding $\tilde{H}_{(u,v)}$. This seems to model joint coupling rather than cascade coupling. The model learns what is common across all layers, but not necessarily how a change propagates from one layer to the next.
4. As a paper in the graph learning domain, why does Equation (9) adopt a Transformer-style attention mechanism instead of using a Graph Attention Network (GAT) [2] approach?

Reference: [1] Zhou, Mingyang, et al. "Modeling personalized retweeting behaviors for multi-stage cascade popularity prediction." Proceedings of IJCAI. 2024.
[2] Veličković, P., Cucurull, G., Casanova, A., Romero, A., Lio, P., & Bengio, Y. (2017). Graph attention networks. arXiv preprint arXiv:1710.10903.

**Questions:**

See weaknesses.

---

### Note · Authors · 2025-11-25

I have read and agree with the venue's withdrawal policy on behalf of myself and my co-authors.